# Kinetically controlled metal-elastomer nanophases for environmentally resilient stretchable electronics

Soosang Chae[1,2,12], Won Jin Choi [3,12] ✉, Lisa Julia Nebel [4], Chang Hee Cho[5], Quinn A. Besford[1], André Knapp[1], Pavlo Makushko[6], Yevhen Zabila[6], Oleksandr Pylypovskyi [6,7], Min Woo Jeong[8], Stanislav Avdoshenko[9], Oliver Sander [4], Denys Makarov [6], Yoon Jang Chung[10], Andreas Fery [1,11], Jin Young Oh [8] ✉ & Tae Il Lee[5] ✉

Nanophase mixtures, leveraging the complementary strengths of each component, are vital for composites to overcome limitations posed by single elemental materials. Among these, metal-elastomer nanophases are particularly important, holding various practical applications for stretchable electronics. However, the methodology and understanding of nanophase mixing metals and elastomers are limited due to difficulties in blending caused by thermodynamic incompatibility. Here, we present a controlled method using kinetics to mix metal atoms with elastomeric chains on the nanoscale. We find that the chain migration flux and metal deposition rate are key factors, allowing the formation of reticular nanophases when kinetically in-phase. Moreover, we observe spontaneous structural evolution, resulting in gyrified structures akin to the human brain. The hybridized gyrified reticular nanophases exhibit strain-invariant metallic electrical conductivity up to 156% areal strain, unparalleled durability in organic solvents and aqueous environments with pH 2–13, and high mechanical robustness, a prerequisite for environmentally resilient devices.

Nanophase mixtures serve as building blocks for functional composites because they can potentially maximize synergistic effects by leveraging the strengths of each component without any compromise[1–4]. However, when combining materials that lack strong mutual interaction, thermodynamically driven phase separation hinders the blending of their chemical and physical properties[5–8].

This challenge is particularly pronounced in composites of metals and elastomers for stretchable conducting membranes[8–12]. Such membranes have long sought after properties that include metal-like conductivity, rubber-like elasticity, mechanical durability, and physicochemical resilience[13–17]. Nevertheless, creating conductive metal-elastomer nanophase mixtures is challenging, as the simple physical

[1]Leibniz-Institut für Polymerforschung Dresden e.V., Institute of Physical Chemistry and Polymer Physics, Hohe Str. 6, 01069 Dresden, Germany. [2]School of Energy Materials and Chemical Engineering, Korea University of Technology and Education, Cheonan 31253, South Korea. [3]Lawrence Livermore National Laboratory, 7000 East Ave., Livermore, CA 94550, USA. [4]Institut für Numerische Mathematik, Technische Universität Dresden, Zellescher Weg 12–14, 01069 Dresden, Germany. [5]Department of Materials Science and Engineering, Gachon University, Seong-nam Gyeonggi 13120, Republic of Korea. [6]Helmholtz-Zentrum Dresden-Rossendorf e.V., Institute of Ion Beam Physics and Materials Research, 01328 Dresden, Germany. [7]Kyiv Academic University, 03142 Kyiv, Ukraine. [8]Department of Chemical Engineering (Integrated Engineering Program), Kyung Hee University, Yongin 17104, Republic of Korea. [9]Leibniz-Institut für Festkörper- und Werkstoffforschung e.V., Institute for Solid State Research, Nothnitzer Str. 49A, 01069 Dresden, Germany. [10]Department of Chemical and Biological Engineering, Korea University, Seoul 02841, Republic of Korea. [11]Technische Universität Dresden, Mommsenstr. 4, 01062 Dresden, Germany. [12]These authors contributed equally: Soosang Chae, Won Jin Choi. ✉e-mail: choi21@llnl.gov; jyoh@khu.ac.kr; t2.lee77@gachon.ac.kr

mixing of metal nanoparticles (or nanowires)[13,14,18] with elastomer precursors or the in-situ reduction of metal nanostructures within the elastomer[19–22] have proven to be insufficient, as they result in either low electrical conductance or reduced mechanical stretchability or elasticity[23]. Alternatively, intrinsically stretchable organic conductors[24,25] or extrinsically configured structures, such as wrinkled/buckled[15,26–28] and kirigami structures[29,30], could solve such performance issues. However, we still expect true metal-elastomer nanophases to be a superior option, especially for usage in high-profit bioelectronic applications which require high-fidelity, long-term stable operation[31] in demanding microcosms within the human body[32]. This is because tailored metal-elastomer nanophases should excel in electrical performance, mechanical durability, and environmental resilience, where each characteristic is inherited from the respective metal and elastomer components. Nanoscale blending forms large interfaces that enhance the adhesion strength between metal and polymer[33–35], as well as "softened" metal nanoparticles and nanostructures result from high surface excess elasticity[36]. Additionally, highly percolated, and adaptable electrical conduction paths[37–39] open up untapped possibilities.

Noble metal deposition on elastomer templates has previously been utilized for stretchable conductors[40–43]. A particularly well-known approach is the thermal evaporation of gold (Au) on styrene-ethylene-butylene-styrene (SEBS) thermoplastic elastomers. During deposition, Au nanoparticles are interpenetrated into the SEBS elastomer and a metal-elastomer biphasic layer is formed, which enables a stretchable metal conductor[40,41]. However, the metallic Au layer is easily cracked under strain, leading to a rather narrow range of electrical and mechanical utilization. In addition, thermoplastic elastomers are vulnerable to heat and organic solvents, which limits their applications in environmentally challenging situations.

Even when nanoblending between immiscible materials is thermodynamically not preferred, kinetics can offer a way to overcome the limits imposed by the physical and chemical properties of materials. In this study, we introduce a kinetically controlled method for forming a well-blended, metal-elastomer nanophase that is energetically unfavorable (Fig. 1). Nanophase, in this context, refers to materials in which two or more different substances are hybridized both physically and chemically, effectively suppressing any degradation in their high performance. This was accomplished by extensively exploring the utilization of the thermal evaporation method for polymeric substrates. While evaporation has been thoroughly investigated for thin film deposition in the past[44], herein we revisited this technique from a kinetic point of view to meticulously mix metal atoms with elastomeric chains.

Vaporized metal atoms are free-form precursors that can be molded into different shapes, from nanoparticles and nanofilms to complex three-dimensional (3D) nanophases. A dramatic variety of structures and phases could emerge when metal atoms are deposited on elastomeric substrates, which can exist in different states ranging from liquid (uncrosslinked) mixtures to solid (fully crosslinked) slabs, depending on their degree of crosslinking. Even seemingly identical solid slabs could have different crosslinking networks and their molecular dynamics could vary drastically as a function of the amount of excess mobile molecules[45–49]. To investigate these phenomena and utilize them for nanoblending, we vaporized Au atoms on polydimethylsiloxane (PDMS) substrates with varying deposition rate, thickness and prepolymer to crosslinker weight ratio, $\varphi$, which is closely related to the migration flux of excessive crosslinkers.

## Results

### Controlling the morphology of Au-PDMS 3D nanophases

These combinatorial methods reveal that the morphology of the resulting nanostructures can indeed be controlled, ranging from nanoparticular, thin film, to complex 3D nanophases, as observed by

optical and transmission electron microscopes (OM and TEM) (Fig. 1b and Suppl. Fig. 1). This structural variety is also evident from the plasmonic color changes due to Au nanostructures (Fig. 1c). What first catches the eye from the images is the distinct formation of nanoparticular phases, where nanoparticles are embedded in PDMS substrates, especially when $\varphi$ is less than 3. This trend showed up almost regardless of the deposition thickness. Note that the deposition thickness, $d$ (nm), mentioned here indicates the thickness measured by the quartz crystal microbalance within the deposition chamber, which can be considered as the product of process time, $t_{process}$ (s), multiplied by the deposition rate ($\text{Å s}^{-1}$). This parameter was used to control the total amount of Au atoms deposited and compare the samples with each other. On the contrary, when $\varphi$ is larger than 10, a thin metal film is formed, consistent with previous reports that utilize standard PDMS substrates for thin film transfer methods[50–53]. For intermediate values of $\varphi$ ($3 < \varphi < 10$), we observe 3D nanostructure phases that show structural variations ranging from elongated nanoparticles and needles to networked reticular complexes, as well as metal densification into PDMS as $\varphi$ increases (Suppl. Fig. 1).

These Au-PDMS nanophases spontaneously form on the exposed surface of PDMS within the vacuum chamber, facilitated by the migration of excessive crosslinkers from the bulk material toward the surface. Inherently then, we should expect the thickness of the PDMS sample also play a role in the nanophase formation process, since it governs the total volume capacity of excessive crosslinkers (see Suppl. Figs. 2 and 3). Modulating the thickness of the PDMS membrane resulted in slight variations in the formation conditions for each nanophase; nevertheless, consistent trends emerged, indicating that increased PDMS thickness led to a higher occurrence of metal encapsulation due to higher migration flux. To gain more insight into the internal structures of these nanophases through the flow of electrons, we measured the electrical conductivity ($\sigma$) for all conditions using a four-point probe method. Based on these measurements, we generated a simplified map (Suppl. Fig. 4) that not only distinguishes between particle, reticular, and thin film phases but also enables estimation of the characteristic lengths within the reticular films, as they exhibit higher values when the Au-PDMS nanophases are more interconnected.

The incorporation of metals into elastomers in kinetically mixed nanophases introduces inherent stress due to the moduli mismatch between the two constituents. Immediately after Au deposition, the Au-PDMS nanophases can be considered to be in a state of mechanical metastability, as they exhibit structural evolution over a relatively extended period of time (~6 h) (Fig. 1d, e, f, and Suppl. Movie 1). We term this evolution process "gyrification" to distinguish it from common wrinkling/buckling. While the term "gyrification" is commonly used to describe the process of forming characteristic folds in the cerebral cortex (Fig. 1i), our investigation reveals a structural similarity, evident by white arrows in the cross-sectional scanning electron microscopy (SEM) image of Fig. 1g, h, and the 3D TEM tomography image of Suppl. Fig. 5 and Suppl. Movie 2. Unlike conventional wrinkles or buckles, which primarily arise in thin metallic films on elastomeric substrates due to external forces such as pre-stretched substrates, and thermal expansion, the gyrified 3D structures in our study originate from internal stress within the blended nanophases and chemical migrations.

The 3D nanophases presented here possess tangential stress gradients, granting them the necessary elasticity, mechanical stiffness comparable to PDMS and malleability for nanoscale folding. In contrast, a conventional bilayer of metal film and PDMS can only experience buckling and lacks the ability to undergo gyrification due to the high energy cost associated with nanoscale creasing of the metal layer. The soft PDMS layer beneath the metal layer is insufficient to induce the necessary stress for folding in the stiffer metal layer. Although the bilayer structure can buckle to form mesoscopic wavy patterns,

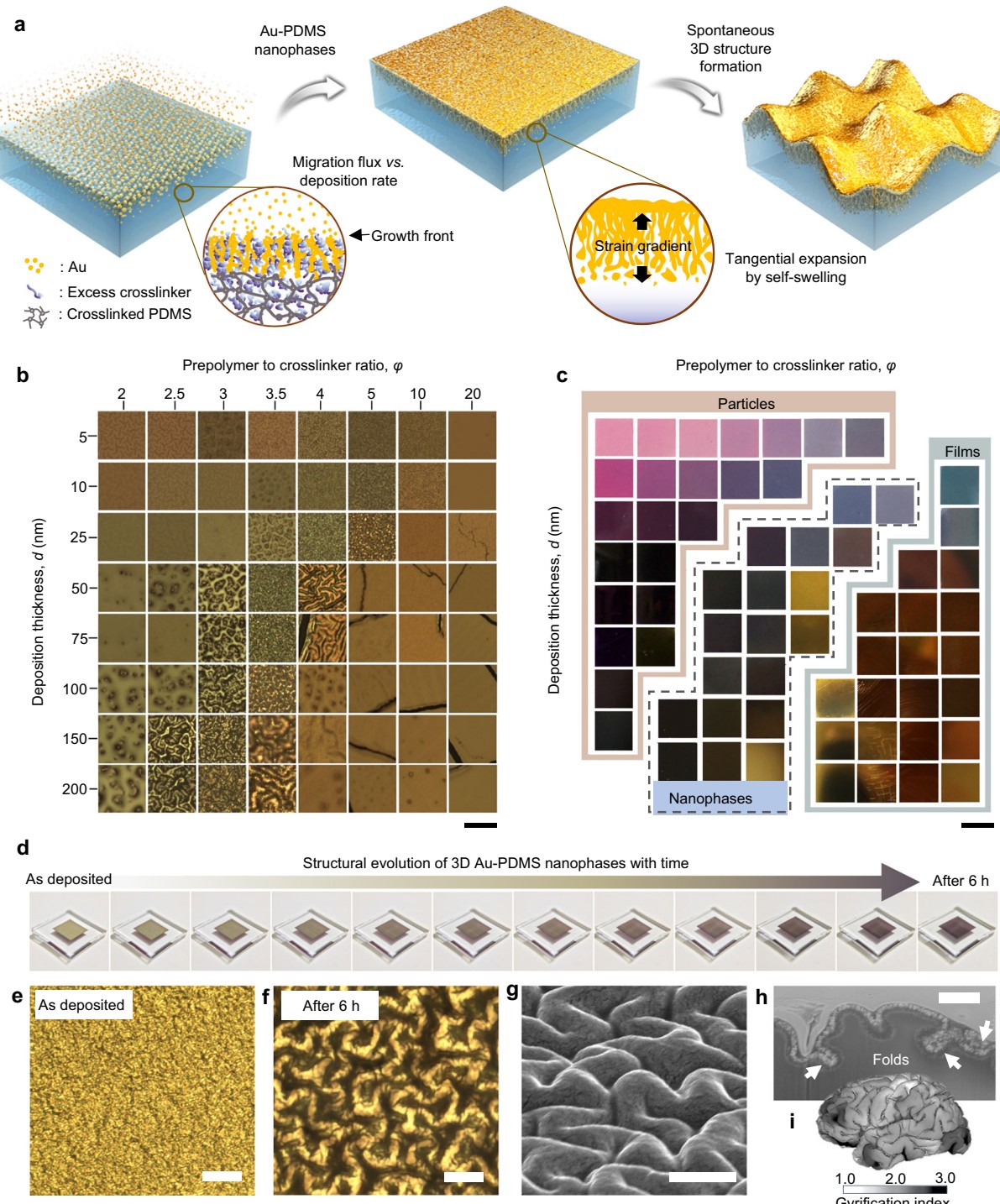

**Fig. 1 | Spontaneously formed metal−elastomer nanophase and microscale 3D structure. a** Schematic illustration of the Au−PDMS nanophase self-forming via vaporized Au deposition on a PDMS substrate. By varying the crosslinking ratio of PDMS and the deposition thickness of metals, we controlled kinetics over three relevant dynamic processes: (1) migration of excess crosslinker, (2) condensation of metal atoms into interlinked Au nanostructures, and (3) growth of nanostructures and formation of the nanophase. Subsequently, spontaneous structural evolution followed to form microscale 3D structures owing to the stress inside. **b** Surface morphology change of Au−PDMS nanophases with various mixing ratio and deposition thickness of the metal. Scale bars denote 150 μm. **c** Photo images showing the corresponding colors of the samples for each condition. Three different of nanostructures were observed: "particles", "3D nanophases", and

"continuous films." Scale bars denote 1 cm. **d** Photographs showing spontaneous structural evolution over a period of 6 h after Au deposition. The length of the PDMS and metal rectangles are 10 mm and 5 mm, respectively. The colors of the samples changed according to 3D structure evolution. **e** An optical microscopy image of as-deposited sample (scale bar is 150 μm). **f** An optical microscopy image of 3D structure-formed (after 6 h) sample (scale bar is 150 μm). **g** Tilted SEM image of gyrified wrinkles (scale bar is 5 μm). **h** A cross-sectional SEM image of the gyrified structure. White arrows indicate folds due to structural expansion (scale bar is 1 μm). **i** An estimated local gyrification index (GI) of the human brain as an example akin to gyrified 3D structures of Au−PDMS nanophases. GI map of the brain adapted with permission from ref. 51. (CC BY 4.0).

achieving folding at the nanoscale is difficult because the stiffness of the metal layer increases exponentially as it deforms. Gyrification can only be observed when the elastic moduli ratio of the two materials, $\overline{E_{PDMS}}/(3E_{metal})$, comprising a composite is high enough to form large amplitude but not too high to have less density. A typical example is the human brain, which can be thought of as a rubber-rubber composite with similar elastic moduli. Here, a similar gyrification process is shown between the Au-PDMS nanopahase and Au layer. Detailed analytical mechanics with equations are provided in Suppl. Note 1, and complementary computational analysis will also be presented to support these observations (see below).

## Growth kinetics of Au-PDMS 3D nanophases

To further investigate the formation kinetics of the Au-PDMS 3D nanophases, we manipulated the flux of Au atoms introduced to PDMS by varying the parameter $\varphi$. This allowed us to observe the interaction and competition between the migration flux of excessive crosslinkers and the deposition flux of Au atoms. Cross-sectional high-resolution TEM (HRTEM) images clearly illustrate that the nanophase morphology is governed by the relative flux difference (Fig. 2a, b). Under fixed conditions of a deposition rate of 2.5 Å s$^{-1}$ and a deposition thickness, $d$, of 100 nm, the nanophase morphology exhibits variations with $\varphi$, transitioning from nanoparticles ($\varphi = 2$), to reticular nanophases ($\varphi = 3.5, 5$), and eventually to thin films ($\varphi = 10$) (Fig. 2a). A similar trend is observed when the situation is reversed, maintaining a fixed $\varphi$ of 3.5 and $d$ of 100 nm but altering the deposition rates (Fig. 2b). Plasmonic color changes stemming from diverse Au nanostructures lend additional support to this observation (Fig. 2c). As the shape of Au nanostructures shifted from nanospheres and nanorods to nanoneedles, the principal localized surface plasmon resonance peaks exhibited a red shift, aligning with the electromagnetic simulation results (Fig. 2d, Suppl. Note 2 and Suppl. Fig. 7). This reaffirms that the resulting nanophase is determined by the relative flux difference rather than the absolute value of individual fluxes.

These results suggest that in terms of growth kinetics, we can classify the nanophases into three distinct cases based on the competition between the excess crosslinker migration flux ($J_{migration}$) and the Au deposition flux ($J_{deposition}$) (Fig. 2e, Suppl. Note 3). The first case occurs when the migration flux significantly exceeds the deposition flux ($J_{migration} \gg J_{deposition}$). Under these conditions, the Au nanoparticles become entirely enclosed by rapidly migrating excessive crosslinkers, preventing further metal coalescence and leading to the formation of particulate nanophases. This phenomenon is akin to sputtering Au atoms onto a liquid substrate to synthesize nanoparticles using silicone oil, as previously reported[54]. In the second case, when the migration rate is comparable to the deposition rate ($J_{migration} \approx J_{deposition}$), crosslinkers are unable to completely cover the initially nucleated Au nanoparticles due to the limited migration time. Consequently, anisotropic growth of Au nanostructures is expected from the exposed areas as seed particles, which is schematically depicted in Suppl. Fig. 8 and confirmed by TEM observations (Suppl. Fig. 9). The third case arises when the deposition rate greatly exceeds the migration rate ($J_{migration} \ll J_{deposition}$), leading to the straightforward deposition of Au as a thin film. Under these conditions, the formation of a nanophase between Au and PDMS is suppressed, enabling the transfer of the metal film, referred to as an elastomeric stamp[51,52].

To establish a correlation between the amounts of excessive crosslinkers and $\varphi$, we also conducted chemical characterizations using $^1$H nuclear magnetic resonance (NMR). Specifically, we performed quantitative analysis on the intensity of peaks corresponding to silicon methyl ($I_4$) and hydride groups ($I_5$), which represent the prepolymer and crosslinkers, respectively (Fig. 2f, g). This approach allowed us to estimate the degree of crosslinking in the PDMS membrane by calculating the ratio of Si−H to Si−CH$_3$ ($I_5/I_4$). The results revealed a clear negative correlation with $\varphi$, indicating an increase in

material conjugation as $\varphi$ increased (Fig. 2h). Our findings align with the mechanism we proposed earlier and inferred from the HRTEM images; smaller $\varphi$ values lead to the formation of nanoparticular phases due to a higher number of excessive crosslinkers, which result in a higher migration flux. Larger $\varphi$ values result in thinner nanophase thickness due to reduced migration, which is also consistent with the measured average value of thickness for samples with different $\varphi$ values, as shown in Fig. 2a, h. Furthermore, the results of a direct contact printing experiment of the PDMS slab using Fourier-transform infrared spectroscopy and atomic force microscopy provide additional support for these findings (Suppl. Fig. 10).

## Mechanism of gyrification in Au-PDMS 3D nanophases

The spontaneous gyrification of reticular nanophases, occurring within a few hours, stands out as one of the most distinctive aspects of our Au-PDMS composite material, setting it apart from many other materials instantly formed from wrinkles/buckles. Nanophases with different $\varphi$ values exhibit significantly different structural evolution timescales and morphologies. For instance, the reticular nanophase ($\varphi = 3.5$) undergoes gyrification, leading to the formation of micrometer-scale deep, large furrows over a period of 6 h (Fig. 3a). In contrast, the particulate nanophase ($\varphi = 2$), initially characterized by fine and smaller (sub-micrometer) structures, develops furrows with micrometer-sized amplitudes within a shorter time frame of 2–3 h (Suppl. Fig. 11). The formation of such microstructures originates from the residual compressive stress within the nanophases. To identify the residual compressive stress accumulated at the interface between Au-PDMS nanophases and bulk PDMS, we conducted spatially resolved Raman spectroscopy on a freshly Au-deposited PDMS membrane with $\varphi = 3.5$ (when the stress was not yet relaxed) and compared the spectra with those obtained from the same sample before deposition (Fig. 3b, c). Following Au deposition, the Raman peaks related to the Si−O−Si stretching of PDMS (centered at 490.8 cm$^{-1}$) clearly showed a red-shift of $\approx 5$ cm$^{-1}$, indicating the possible occurrence of spatial strain in the PDMS adjacent to the Au layer.

The Raman data are consistent with the results obtained from finite element method (FEM) simulations based on hyper-elastic material models (Suppl. Note 4 and 5). Specifically, we used a hyperelastic Mooney-Rivlin material with parameters derived from uniaxial tensile tests (see Suppl. Fig. 12 and Suppl. Table 1 for the uniaxial tensile tests for the $\varphi = 3.5$ PDMS). For the plain Au layer and the Au-PDMS layer (an elastic rectangular block of dimensions of 24 μm × 24 μm × 20 μm in x-, y- and z-direction), we used two geometrically exact Cosserat shells as described in the literature[55]. To model the swelling ratio, we attached a swollen plain Au layer and a swollen Au-PDMS layer to a stretched-out plain PDMS layer. After releasing the stretch on the plain PDMS layer, a 3D microstructure formed due to the stress mismatch. For all simulations, we chose the same maximum penetration depth of Au atoms as in the HRTEM images (Fig. 2a). There was almost no dependence on the deposition thickness in the case of $\varphi = 3.5$, which corresponds to our experiments.

In our computational analysis, we explored the surface morphology using three key parameters: Young's modulus ($E_Y$), the swelling ratio of the Au-PDMS nanophase layer ($Q$), and the thickness of the top plain Au layer ($t_{Au}$). Among the various experimental conditions, in particular we focused on the effect of deposition thickness on the swelling and releasing behavior of gyrified structures. In our simulations, we assumed that that with an increase in the process time, $t_{process}$, at a fixed deposition rate, both the swelling ratio of the Au-PDMS nanophase layer, $Q$, and Young's modulus, $E_Y$, also increase (Fig. 3d). The calculated surface morphology after stress release closely matched the experimental data in many aspects, including the shape, feature size, and general trend with increasing $t_{process}$. The transition from hexagonally arranged islands to a labyrinth pattern was

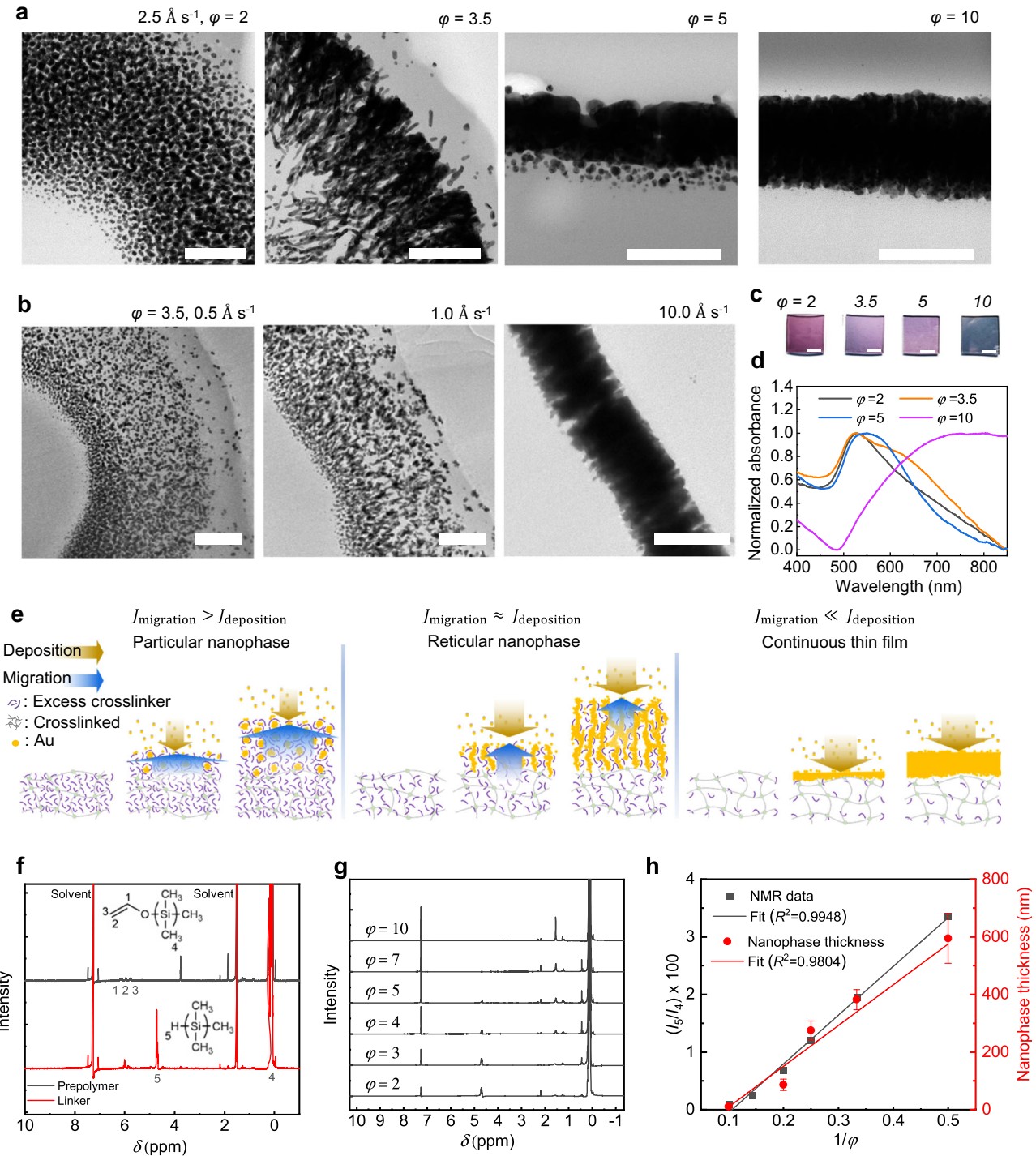

**Fig. 2 | Growth Au-PDMS nanophases controlled by chemical kinetics.** Cross-sectional HRTEM images of Au-PDMS membranes with (**a**), various prepolymer to crosslinker weight ratio of PDMS, $\varphi = 2, 3.5, 5,$ and 10 at a deposition rate of 2.5 Å s$^{-1}$, and with (**b**), various deposition rates of 0.5, 1.0, and 10.0 Å s$^{-1}$ at $\varphi = 3.5$, displaying a variation in Au-PDMS nanophases depending on which kinetics is more dominant during the deposition (polymer migration versus rate of metal deposition). Especially, the reticular nanophase formed at $\varphi = 3.5$ at a deposition rate of 2.5 Å s$^{-1}$ is an interpenetrated Au–PDMS nanophase. Scale bars denote 200 nm. **c** Photo images of the samples (scale bars denote 3 mm). **d** Absorbance spectra of 10 nm Au-deposited samples at 2.5 Å s$^{-1}$ on PDMS membranes with different crosslinker concentrations ($\varphi = 2, 3.5, 5, 10$). **e** A plausible nanophase formation mechanism. **f** $^1$H NMR spectrum of neat PDMS prepolymer and curing agent. **g** $^1$H NMR spectrum of CDCl$_3$, which was incubated with PDMS at different mixing ratios of the two components. **h** A negative correlation was observed between the peak intensity ratio of Si–H to Si–CH$_3$ from the 1H NMR spectrum (**g**) and $1/\varphi$, indicating a direct relationship between chain migration and the resulting nanophase morphology. Error bars indicate the standard deviation of three independent measurements.

confirmed when $t_{Au}$ is 80 nm, $Q$ is 1.3, and $E_Y$ is 10 GPa, aligning well with observations in the OM images. These results support our hypothesis that the spontaneous evolution of the 3D structure is initially driven by the compressive stresses of an Au-PDMS nanophase accumulated by excessive crosslinkers. However, it is important to note that the complete replication of gyrified experimental results with deep folds in simulations was challenging using this simplified FEM model, even when scanning all possible computational parameters.

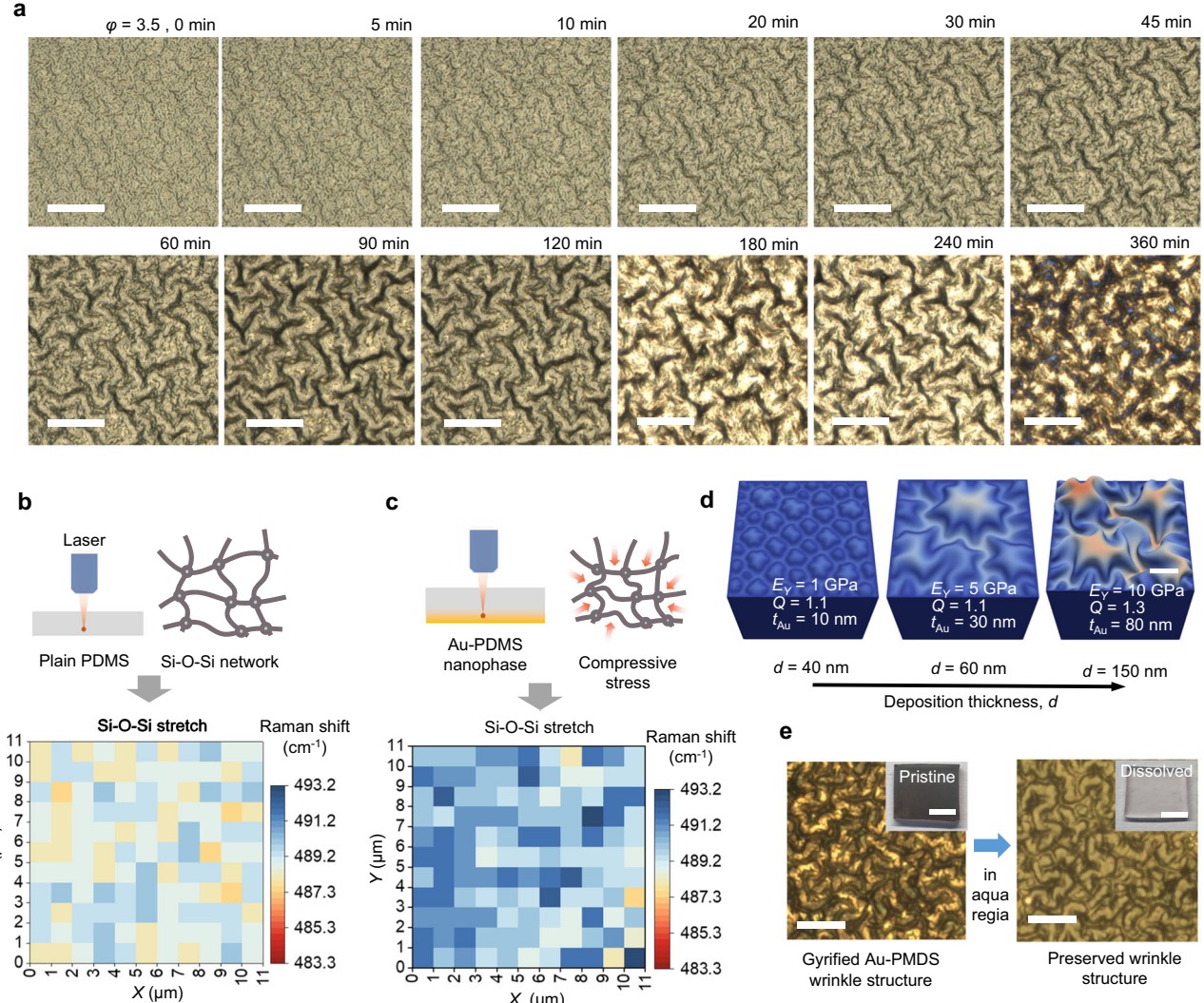

**Fig. 3 | Gyrification of Au-PDMS reticular nanophase. a** Optical microscopy images showing the spontaneous gyrification of the Au-PDMS reticular nanophase after Au deposition over a period of 6 h. Scale bars in the photographs denote 15 μm. **b** Raman mapping image of plain PDMS. **c** Raman mapping image of as-Au–deposited substrate showing residual stress formed during the reticular growth of Au nanostructures inside the vacuum chamber. **d** Finite element method (FEM) simulation of morphology features under different mechanical parameters and Au deposition thicknesses. Scale bar denotes 10 μm. **e** Optical microscope images of a gyrified nanophase sample before and after dissolving Au in aqua regia. Scale bars denote 15 μm. Inset photo images represent apparent color change according to dissolving Au. Scale bars of the insets denote 1 cm.

This is mainly because gyrification cannot be solely induced by compressive stresses with fixed force; rather, it involves a complex dynamic process that includes chemical migrations and changes in volume and compressive force vector over time.

Based on real-time OM observations (Suppl. Movie 1), together with HRTEM morphologies and FEM simulations, we conclude that the gyrification phenomenon is initially driven by accumulated compressive stress. However, it is further influenced by the subsequent self-swelling of uncross-linked, low-molecular-weight PDMS oligomers for a couple of hours. Macroscale expansion due to the self-swelling of oligomers was observed and quantified by measuring the change in a patterned deposition area (Suppl. Figs. 13 and 14). The different nanophases exhibited variations in expansion: the reticular nanophase displayed a roughly 10% change in measured area, while the particulate nanophase showed a change in area of approximately 3% (measured by the photolithographically defined gap between before and after gyrification, Suppl. Fig. 14). Another strong piece of evidence suggesting the self-swelling of PDMS oligomers can be found by etching out metal layers after gyrification. If the nanostructures collapse after the etching process, it indicates that there were no crosslinked PDMS nanostructures but only Au nanostructures with excessive crosslinkers. However, if they maintain their gyrified morphologies even after the etching process of Au, it indicates that the PDMS oligomers themselves migrated and crosslinked to form the nanophase structures. It was observed that the gyrified PDMS structures were maintained even without the Au layer (Fig. 3e), strongly suggesting the self-swelling of oligomers as well as their crosslinking into reticular nanophases during the gyrification processes.

## Elastomer-like stretchability with metal-like conductivity

As mentioned earlier, the primary goal of forming a metal-elastomer nanophase is to achieve pure PDMS stretchability while maintaining Au electrical conductivity. We argue that our Au-PDMS nanophase materials demonstrate such characteristics. In a sample with optimized parameters (1.25 mm thick PDMS membrane, $\varphi = 3.5$, Au deposition rate of 2.5 Å s$^{-1}$, and $d$ of 100 nm), we observe a conductivity as high as $1.4 \times 10^4$ S cm$^{-1}$ (Fig. 4). This initial conductivity obtained by four-point probe measurements did not change upon the application of uniaxial

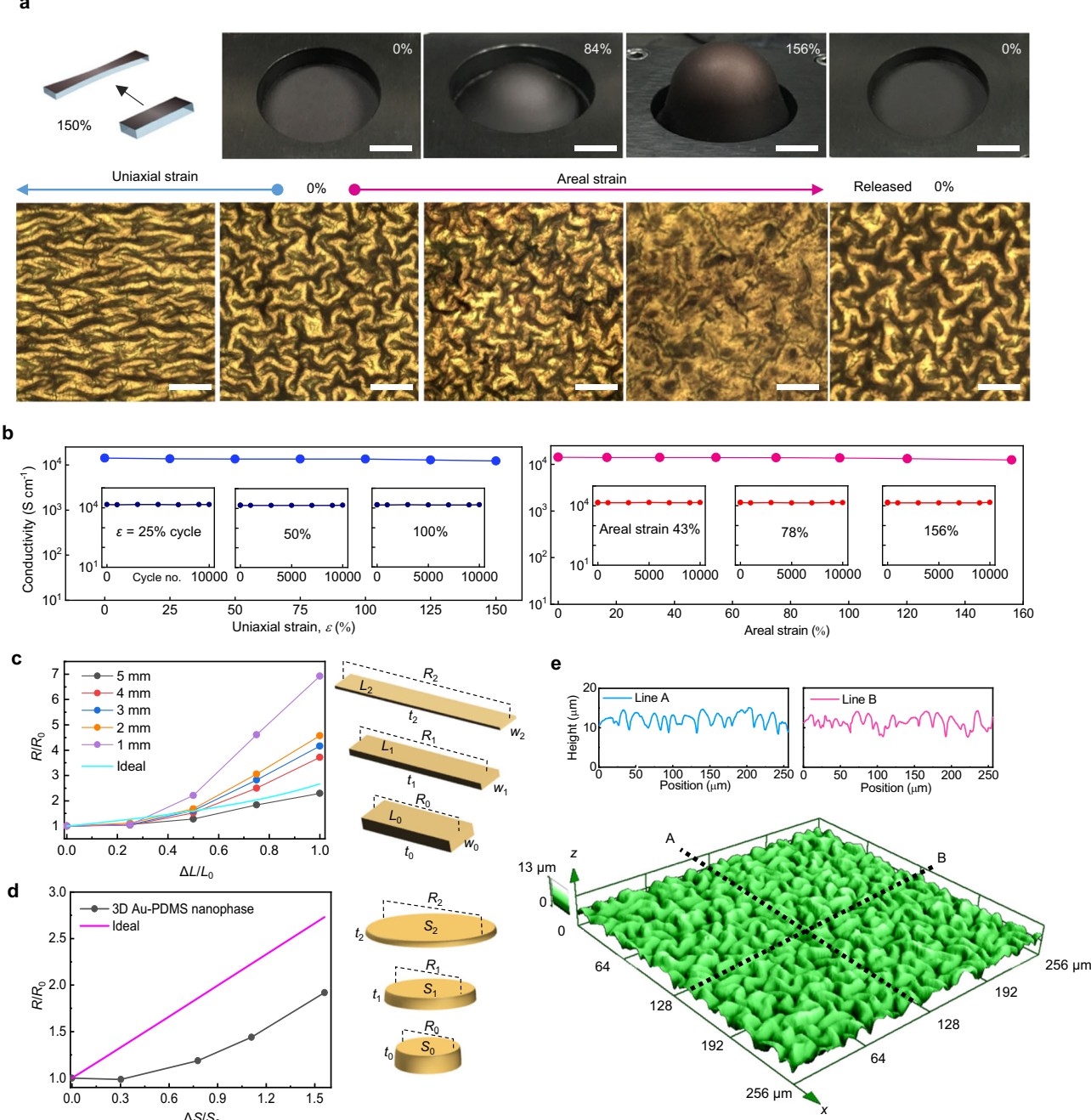

**Fig. 4 | Highly strain-invariant electrical conductivity achieved by the gyrified 3D structure. a** Optical microscopy (lower) and photo images (upper) showing stress dissipation by the gyrified 3D structure coupled with internal Au–PDMS nanophase upon applied strain both at uniaxial (leftwards) and areal (rightwards) strains. Strain-released images (0% at the outermost right side) show reversibility. Scale bars in the photo images denote 1 cm. Scale bar denotes in the optical microscopy images 150 μm. **b** Electrical conductivity of >$10^4$ S cm$^{-1}$ up to uniaxial 150% (left) and areal 156% (right) strain, along with cyclic stability over 10,000 cycles of each strain (insets). **c** A uniaxial strain-dependent relative resistance changes of the 3D structure of Au–PDMS nanophase strap pads with various initial widths (5, 4, 3, 2, and 1 mm). A cyan-colored ideal curve represents calculated relative resistance change. **d** An areal strain strain-dependent relative resistance changes of the 3D structure of Au–PDMS nanophase disc-shaped pad. A magenta-colored ideal curve represents calculated relative resistance change. **e** Confocal microscopy scan image of the 3D structure of Au–PDMS nanophase sample. Line profiles extracted at the dot line A and B show that the high specific surface area contributed to the strain-invariant conductivity.

strain up to 150% (Fig. 4a, b, blue) or areal strain up to 156% (Fig. 4a, b, red). No mechanical failures, such as cracks or delamination, were observed during the stretching tests, confirming its rubber-like elasticity. Moreover, the initial conductivity remained constant even after 10,000 stretching–releasing cycles with uniaxial and areal strains in the same range (Fig. 4b). Additionally, it exhibits superior long-term

stability, as the electrical performance remained virtually constant for more than 8 years of shelf life (Suppl. Fig. 15).

Depending on the application, electrodes can generally be classified into two types: active and passive. Active electrodes are used to directly inject or extract currents into/from a system, while passive electrodes serve to carry currents between different components or

parts of a circuit. Among these, strain-invariant electrical properties are especially important for passive bus lines, ensuring a stable and reliable conductive path within and between circuit elements. To address this importance and cater to practical applications, we studied the relative resistance of our 3D Au-PDMS nanophase samples by employing two-terminal measurements with electrode widths ranging from 1 to 5 mm (Fig. 4c).

The standout feature of our Au-PDMS nanophase is its invariance to strain, greatly surpassing other state-of-the-art materials (Suppl. Tables 2, 3, Suppl. Fig. 16)[13–15,24,25,56]. The "ideal elastic conductor" in Fig. 4c depicts a hypothetical sample with strain-invariant resistivity in two-terminal measurements, only portraying the geometric change in resistance that would come with stretching, including considerations regarding Poisson's ratio (Suppl. Fig. 17). Under this condition, the calculated resistance increases by 2.5 times at 100% uniaxial strain (calculation details in Suppl. Note 6). Comparing this to the measured resistance changes of our Au-PDMS, we arrive at a compelling result: even up to 30% uniaxial strain, the resistance changes of our samples were significantly lower than those of an ideal elastic conductor. This intriguing behavior comes from the unfolding of the 3D gyrified structures upon stretching, effectively compensating for the increased resistance due to dimensional changes.

The experimental and calculated resistance changes under areal strain showed the same tendency, where the Au−PDMS 3D nanophase samples exhibited a smaller resistance change than an "ideal elastic conductor" with up to 156% areal strain (Fig. 4d, Suppl. Fig. 18). Note that areal strain, in this context, is different from biaxial strain, as areal strain involves 3D stretching using a sphere, while biaxial strain is

limited to stretching within a 2D plane (Suppl. Table 3). The unprecedented characteristics of our Au-PDMS nanophase material is primarily attributed to both the inherent stretchability of the reticular nanophases as well as the auxetic nature of the extrinsically gyrified 3D morphology (Suppl. Fig. 19). The combination of these intrinsic and extrinsic properties provides the high adhesivity and the structural, electrical elasticity required to prevent delamination and deterioration of electrical conductance in composite materials. Our view is further supported by the high gyrification index of 1.75 calculated in our samples, which is defined as the ratio of the actual surface area to the projection area (Fig. 4e).

### Environmental durability for soft electronics

Following the electrical and mechanical properties of our gyrified reticular Au-PDMS nanophase samples, we showcase their multifaceted performance for applications in environmentally resilient soft electronics. Our samples display excellent stability against pH, chemical, and thermal exposure, as well as mechanical abrasion (Fig. 5a and Suppl. Note 7). After immersion in various polar and non-polar solvents (water, ethanol, acetone, chlorobenzene, and toluene) for 1 day, the conductivity of the nanophase samples slightly decreased, especially in organic solvents. This is possibly because of the deformed conduction network that comes from the swelling of the PDMS matrix itself. However, after the nanophases were dried, the conductivity almost fully recovered to within a 5% margin of error. Meanwhile, the conductivity was preserved under harsh pH conditions (pH 2–13), demonstrating this material's suitability for bioelectronics applications in the stomach (pH 1.3–3.5) as well as for use on skin wounds (pH

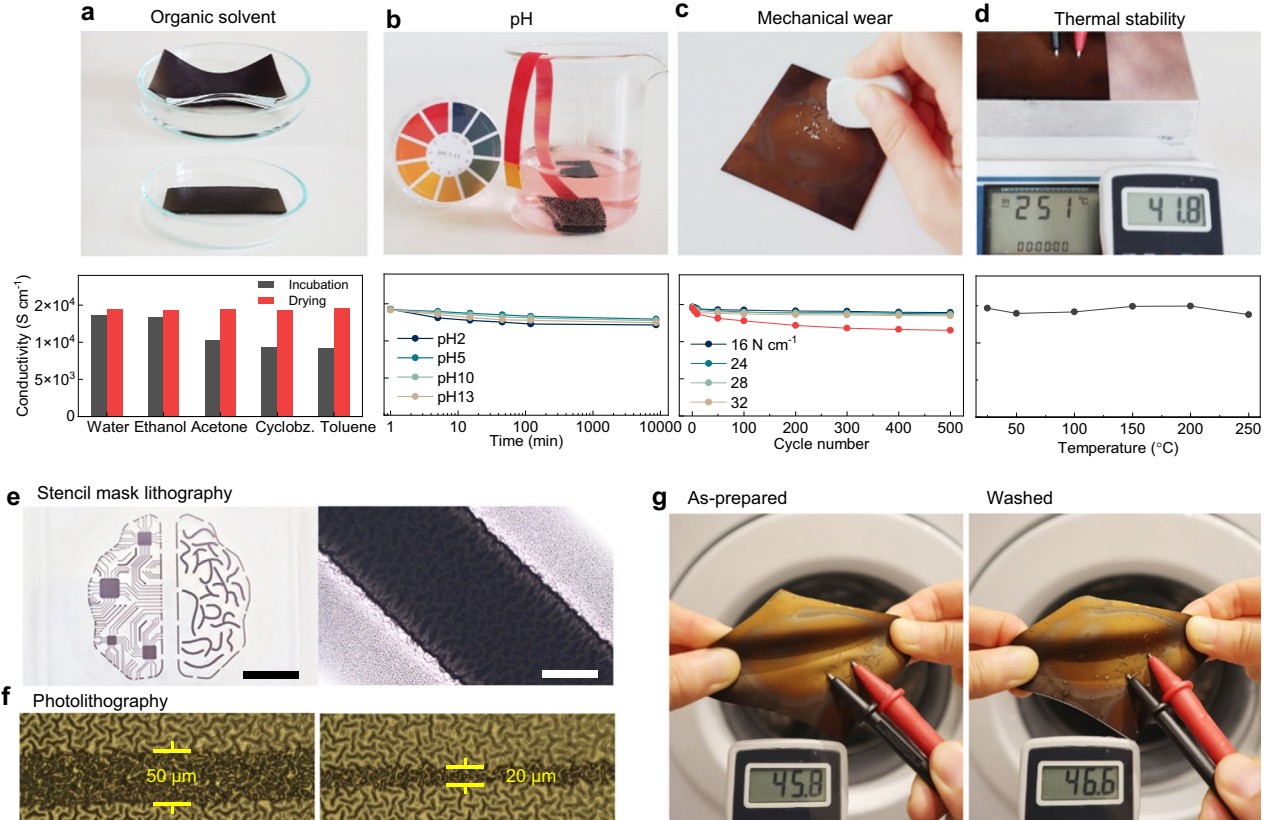

**Fig. 5 | Multimodal stability of the 3D structure of Au−PDMS nanophases.**
**a** Durability performance of the 3D structure of Au−PDMS nanophases in various organic solvents (cyclobz. stands for cyclobenzene), (**b**) pH (pH 2–13), (**c**) eraser and adhesive tape test (red line from eraser test and other lines from tape test), and (**d**) thermal annealing up to 250 °C. **e** Patterning of the 3D structure of Au−PDMS nanophases using stencil mask lithography (black color scale bar is 1 cm and white color scale bar is 50 μm) and (**f**) conventional photolithography, which can be realized for patterning of stretchable electronics. **g** Washing test, displaying no degradation of electrical function under machine washing with detergent, as shown in Suppl. Movie 5.

7.2–8.9) (Fig. 5b). The conductivity of the gyrified Au–PDMS reticular nanophases was also almost fully maintained during adhesive tape tests (ranging from 16 to 32 N cm⁻¹) for 500 cycles (Fig. 5c). Even after an abrasion test using an eraser (Suppl. Fig. 20 and Suppl. Movie 3), the conductivity remained in the same order of magnitude (red in Fig. 5c). In contrast to other thermoplastic elastomer-based stretchable electrodes, the gyrified Au-PDMS nanophases also showed excellent thermal stability up to 250 °C for 2 h (Fig. 5d). This expands their potential use in interfacing with functional elements that require thermal annealing during fabrication and operation at elevated temperatures, such as thermoelectric and heater modules, or during hydrothermal sterilization processes (Suppl. Movie 4, and Suppl. Fig. 21).

Furthermore, we also verified the reliability of the standard lithography processes for the gyrified Au-PDMS nanophase (Fig. 5e, f). The chemical (organic solvents) and thermal (annealing processes for photoresists) durability of the Au–PDMS nanophase samples enabled microscale patterning down to a 20 µm feature size using conventional photolithography with complex circuit designs (Suppl. Figs. 22, 23). The Au–PDMS nanophase samples were also found to be washable in a laundry machine. The initial conductivity and stretchability showed almost no change even after 20 cycles of 15 min. washing with detergent (Fig. 5g, Suppl. Fig. 24, Suppl. Movie 5). These findings imply that this stretchable conducting membrane can be used to produce lithographically defined, functional devices intended for repeated use with reliable operation in the field of wearable electronics for virtual reality (Suppl. Note 8, Suppl. Figs. 25, 26, and Suppl. Movie 6), including educational and medical applications.

Our 3D structure of Au-PDMS nanophases holds considerable appeal as a stretchable conducting membrane for soft and bioelectronics applications and may find extensive use in various fields, including biomonitoring within the digestive tract, multimodal implantable devices like ocular prostheses, soft robotics employing mechanical strain-gated logic gates, and functional fabrics for space exploration.

## Methods

### Kinetically controlled metal–elastomer nanophases

For the preparation of PDMS membranes, a mixture of two components, prepolymer PDMS and linker PDMS (Sylgard 184, Dow Corning), was blended with varying ratios ($\varphi = 2\text{--}20$). The mixture was subsequently thermally cured at 80 °C for 4 h. The PDMS membranes were affixed onto sample holders within a physical vapor deposition chamber manufactured by iNFOVION. To prevent any potential evaporation of small molecules within the PDMS slab, we maintained a consistent vacuum pressure for all samples. Gold (Au, 99.9999%) was deposited using thermal evaporation at room temperature, with a constant base pressure of $1.0 \times 10^{-6}$ Torr. The in-situ deposition rate and amount were monitored using a quartz crystal microbalance. The deposition rate of Au was maintained at 2.5 Å s⁻¹ for all samples, except for samples with varying deposition rates (ranging from 10 Å s⁻¹ to 0.5 Å s⁻¹). A pre-deposition step was conducted for 1 min at a constant rate of 2.5 Å s⁻¹, with the shutter closed. The main deposition phase commenced by opening the shutter and continued for 400 s, resulting in the formation of a nanophase that exhibited optimal stretchable electrode performance. Following deposition, the samples were promptly removed from the chamber and allowed to age for 6 h in ambient air. Subsequent characterization and analysis were performed without any additional surface treatments.

According to the materials safety data sheets (MSDS) for the prepolymer PDMS, it includes 0.5 wt% xylene, 0.2 wt% ethylbenzene, >60 wt% dimethylvinyl-terminated dimethyl siloxane, 30–60 wt% dimethylvinylated and trimethylated silica, and 1–5 wt% tetra(-trimethylsiloxy) silane. Meanwhile, the MSDS for the Sylgard 184 "Curing Agent" indicates 0.19 wt% xylene, <0.1 wt% ethylbenzene, 55–75 wt% dimethyl, methyl hydrogen siloxane, 15–35 wt% dimethylvinyl-terminated dimethyl siloxane, 10–30 wt% dimethylvinylated and trimethylated silica, and 1–5 wt% tetramethyl tetravinyl cyclotetrasiloxane[57]. Additionally, a siloxane complex containing a few wt% Pt may be present in the curing agent as a catalyst, which we believe plays the most important role in crosslinking. However, the exact concentrations are unknown. Given these conditions, we believe that most experimentalists would agree that core components of prepolymer and crosslinker, would include difunctional vinyl-terminated PDMS chain as the prepolymer, where crosslinking occurs via the SiH-containing, a low molecular weight trimethylsiloxy-terminated PDMS with platinum catalysts assisting the formation of a siloxane cross-linking network[58].

The viscosity of the prepolymer and the mixture immediately after mixing with the curing agent is 5500 mPa s and 4000 mPa s, respectively. For reference, the molecular weight of Sylgard 184 has been reported to be about $M_n \approx 4500$ g mol⁻¹ and $M_w \approx 19,000$ g mol⁻¹ (PI = 4.24) in literature[59]. $M_n$ and $M_w$ denote the number-averaged and weight-averaged molecular weight, respectively, and PI is the polydispersity index.

### Stretching performance, mechanical and optical characterization

To uniformly apply uniaxial and areal strain to the 3D structure of Au-PDMS nanophases samples, a custom-made 1-D and 2-D screw-based stretching apparatus was integrated with an optical microscope (BX-51P, Olympus), confocal microscope, and a four-point probe station (Loresta-GP MCP-T600, MITSUBISHI CHEMICAL) for electrical property measurements. The strain rate used for the strain-dependent conductivity measurements was set at 10% s⁻¹. Cyclic stretchability, tested up to 10,000 cycles, was assessed using a stretching machine (Flexible Materials Tester, Hansung Systems, Inc.). For viscoelasticity measurements, dynamic mechanical analysis (DMA) was conducted using a TA Instrument Q800 instrument with a gas cooling accessory and a fiber/film tension modulus. Young's modulus of both plain PDMS and the 3D structure of Au-PDMS nanophases samples was determined using a MultiXtens II HP instrument from Zwick, following the DIN 53504/S2/30 protocol. The gyrification index was determined from the data acquired through laser scanning confocal microscopy (LSM, Olympus LEXT OLS4100 laser scanning digital microscope). UV–Visible spectra of the samples were obtained using V-770 UV-Visible/NIR spectrophotometer. Raman mapping was obtained from Raman spectra (LabRam Aramis, Horiba Jovin Yvon) of the samples in 10 µm × 10 µm area.

### TEM characterization

For the cross-sectional TEM images, PDMS samples were prepared as lamellar thin slices using focused ion beam (FIB) techniques with the JIB-4601F instrument from JEOL and the Quanta 3D FEG instrument from FEI. Subsequent TEM analysis was performed using a JEM-ARM 200F microscope from JEOL. To generate the 3D TEM tomography image of the Au-PDMS nanophases in the sample, a series of tilted TEM images (30 images in total) were acquired by rotating the sample holder from −21° to 66°. This process was carried out at a magnification of 50,000 with an exposure time of 0.5 s using a CCD camera (Orius SC200D, Gatan) mounted on an Ultra Corrected Energy Filtering Transmission Electron Microscope (UC-EF-TEM) system (Libra 200 MC TEM, Carl Zeiss). The acquired images were then subjected to an image reconstruction process, specifically Filtered Back Projection (TEMography), to generate the final 3D image.

### NMR characterization

The modulation of non-covalently tethered PDMS molecules within the PDMS matrix can be achieved by adjusting the prepolymer to crosslinker ratio. To characterize the residual solvent within the PDMS

matrix, a series of free-standing PDMS membranes were immersed in CDCl₃ for ¹H nuclear magnetic resonance (NMR) analysis, allowing us to track the presence of uncrosslinked PDMS over time. Freshly cured PDMS membranes, prepared with varying $\varphi$ values, were sectioned into $4 \times 5 \times 10$ mm³ pieces and placed in an air-tight vessel containing 1.2 mL of CDCl₃ (Eurisotop) for overnight incubation. Due to higher adsorption of the solvent by PDMS membranes with lower $\varphi$, an additional 0.8 mL of CDCl₃ was added after 60 min. Subsequently, a portion of each sample's supernatant was collected for ¹H NMR measurements (500.13 MHz) using an AVANCE III 500 Spectrometer (Bruker, Germany) at a temperature of 30 °C.

## Micropatterning for integrated circuit

The photolithography of the Au-PDMS nanophase samples was conducted following standard procedures. Specifically, the sample was spin-coated with a positive photoresist (positive AZ 5214E®, MicroChemicals) and then subjected to a soft bake at 110 °C for 2 min. Subsequently, the sample was exposed to a 365-nm UV light source (KLS-100H-LS-150P, DONGWOO Optron) using a patterned photomask. After exposure, the sample was immersed in a developer solution (AZ® 327, MicroChemicals) with vigorous shaking for 1 min. A post-bake step at 190 °C for 10 min was performed to ensure complete curing of the residual photoresist. Selective etching of the nanophase was achieved by incubating it in aqua regia, a mixture of HCl and HNO₃ in a 9:1 volume ratio. Any remaining photoresist was removed using a universal photoresist stripper (AZ® 100, MicroChemicals) for lift-off. For stencil mask lithography, a metal stencil mask (0.1 mm thick stainless steel, Devora Electronics) was placed directly on a plain PDMS membrane ($\varphi = 3.5$, thickness 0.65 mm) without a spacer.

## Environmental resilience test

To assess chemical stability, the as-fabricated 3D structure of Au-PDMS nanophase samples were immersed in various solvents (water, ethanol, acetone, c-Benzene, and toluene) overnight, followed by conductivity measurements (marked as "incubation" in the "Organic solvent" section of Fig. 5a). Subsequently, the samples were deswelled at 60 °C overnight, and the conductivity was measured again (marked as "drying" in the "Organic solvent" section of Fig. 5a). For pH stability testing, solutions of HCl and KOH were used to create different pH conditions. The as-fabricated samples were then immersed in the respective pH solutions for varying durations (up to 10,000 min). In the tape adhesion test, a strong adhesive tape (3 M) was repetitively attached and detached from the surface of the 3D structure of Au-PDMS nanophases for 500 cycles. The conductivity change was measured after each attachment cycle. In the eraser test, the surface of the as-fabricated Au-PDMS nanophases was repeatedly rubbed with a pencil eraser (TOMBOW®) for a specified number of cycles. The rubbing strength was determined by the specific contact area (2 cm × 2 cm) and the applied push force, which was ~0.245 kPa, a force sufficient to generate eraser dust. This procedure was recorded in Suppl. Fig. 20 and Suppl. Movie 3. Thermal stability trials were performed by heating the samples on a hot plate (SMHS-3, DAIHAN). The conductivity of the samples was evaluated after annealing at various temperatures. For the laundry test, the sample (without encapsulation) was subjected to washing both with and without detergent (Persil Gel Detergent, Germany) in a washing machine (WW70T4042EE/EG, Samsung) for 15 min at room temperature. This washing procedure was repeated 20 times for the cyclic washing test.

## Data availability

The data that support the findings of this study are available from the corresponding authors upon request.

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

## Acknowledgements

This research was supported by the National Research Foundation of Korea (NRF) (2021R1C1C1009925, 2020R1A6A1A03048004, and 2019R1A6C101052) and funded by the Korea Evaluation Institute of Industrial Technology (KEIT) (20015898, 20012710 and 20019105). This work was also supported by the Korea Institute for Advanced Technology (KIAT), the Ministry of Trade, Industry and Energy (MOTIE) of the Republic of Korea (nos. P0017363), and the GRRC program of Gyeonggi province (GRRCKYUNGHEE2023-B03). We thank the Korea Basic Science Institute (KBSI) for the technical support. We gratefully acknowl-edge Anik Kumar Ghosh, Uta Reuter, and Petr Formánek for assistance with FEM simulations, FIB, and TEM analysis, respectively. We thank Jeeho Park, Minji Cho, and Hyemin Park for assistance with image pro-cessing, as well as Jonas Schubert and Duwon Jeong for fruitful dis-cussions. A.F. and S.C. acknowledge support from Deutsche Forschungsgemeinschaft (DFG, German Research Foundation) within SPP 2100, project number 359715917; A.F., A.K., O.S., and L.J.N. acknowledge support from Deutsche Forschungsgemeinschaft (DFG) within project number 386450667. This research was also performed within project LaSensA under the M-ERA.NET scheme and was funded by the Saxon State Ministry for Science, Culture, and Tourism (Germany) and co-financed with tax funds on the basis of the budget passed by the

Saxon state parliament and funded by the Deutsche Forschungsgemeinschaft (DFG)- CRC–1415 - 417590517. S.C. and A.F. acknowledge support from the German Science Foundation with SPP 2100, project number 404941515. German Research Foundation grant 600/20–1 640690U12AB123456 (L.J.N.). This work is financially supported in part by German Research Foundation (DFG) grants MA 5144/13–1, MA 5144/28–1, European Commission HORIZON RIA (project REGO; ID: 101070066) and Helmholtz Association of German Research Centres in the frame of the Helmholtz Innovation Lab "FlexiSens". This work was performed under the auspices of the U.S. Department of Energy by Lawrence Livermore National Laboratory under Contract DE-AC52-07NA27344. W.J.C. gratefully acknowledges the LLNL LDRD Programs for funding support of this project under No.22-ERD-056 and 24-LW-035. This work was also supported by a Korea University grant.

## Author contributions

J.Y.O., S.C., W.J.C., and T.I.L. contributed to conceptualization. J.Y.O., S.C., Q.A.B., L.J.N., A.K., C.H.C., P.M., Y.Z., S.A., O.P., and T.I.L. were involved in developing and defining the methodology. S.C., J.Y.O., W.J.C., D.M., Q.A.B., L.J.N., A.K., C.H.C., P.M., Y.Z., S.A., O.P., and T.I.L. conducted the investigation. S.C., J.Y.O., Q.A.B., L.J.N., C.H.C., P.M., Y.Z., S.A., O.P., T.I.L., and M.W.J. handled visualization tasks. J.Y.O., D.M., T.I.L., W.J.C., and A.F. led funding acquisition efforts. J.Y.O., D.M., T.I.L., W.J.C., and A.F. managed the project administration. J.Y.O., D.M., W.J.C., A.F., O.S., S.A., and T.I.L. supervised the project. J.Y.O., S.C., W.J.C., D.M., Q.A.B., Y.J.C., and T.I.L. participated in writing the manuscript.

## Competing interests

The authors declare no competing interests.
