## [Peer Review File · Nature Communications]

Kinetically controlled metal-elastomer nanophases for environmentally resilient stretchable electronicsEditorial Note: This manuscript has been previously reviewed at another journal that is not operating a transparent peer review scheme. This document only contains reviewer comments and rebuttal letters for versions considered at *Nature Communications* .

REVIEWERS' COMMENTS

Reviewer #1 (Remarks to the Author):

Reviewer report on NCOMMS-23-58296-T

"Kinetically Controlled Gold–Elastomer Nanophases for Environmentally Resilient Stretchable Electronics" by Soosang Chae, Won Jin Choi, Lisa Julia Nebel, Chang Hee Cho, Quinn A. Besford, André Knapp, Pavlo Makushko, Yevhen Zabyla, Oleksandr Pylypovskyi, Min Woo Jeong, Stanislav Avdoshenko, Oliver Sander, Denys Makarov, Yoon Jang Chung, Andreas Fery, Jin Young Oh and Tae Il Lee.

I have carefully read the revised version of the manuscript NCHEM-23081735 transferred to Nature Communications. I am fully satisfied with the response of the authors to the points addressed in my earlier Reviewer Report to this manuscript and the changes made in the revised manuscript. I think that these changes have improved the readability and clarity of the manuscript. Hence, I recommend publication of the revised version of the manuscript in Nature Communications.

Reviewer #2 (Remarks to the Author):

The authors report on a method to obtain strain-invariant stretchable conductors by very slow evaporation of Au onto PDMS containing an excess of mobile crosslinker molecules. Based on extensive and careful analytical investigations and complementary simulations they developed a model for the reproducible formation of a buckled three-dimensional nanostructure which combines remarkable stretchability, strain-invariance, and robustness under harsh environmental conditions. Key control parameters are the metal deposition rate and the amount of crosslinker.

Stretchable conductors are of strong interest, particularly for stretchable electronics and soft robotics, and have been investigated extensively. Some studies also report strain-invariant stretchable conductors. This feature is crucial in some applications. Nevertheless, the present work, from my point of view warrants publication in a high-impact journal like Nature Communications because, the reported approach is novel and, different from other approaches, it achieves the desired properties like high stretchability, strain-invariance, and resistance to harsh environment at the same time.

The manuscript is well written and structured, and the comments of the reviewers have been adequately addressed. Concerning earlier work, I suggest to include two additional papers in the introductory part. In the paper: "Highly Deformable Nanostructured Elastomeric Electrodes With Improving Conductivity Upon Cyclical Stretching" G. Corbelli et al., first published: 29 August 2011, <https://doi.org/10.1002/adma.201102463>, the group of Paolo Milani reports on stretchable conductors based on Au nanostructures obtained by deposition of Au nanoclusters onto PDMS. This approach, which was refined in various follow-up papers, is different from the present one but should be mentioned in the introduction because the authors also report on Au/PDMS nanostructures for stretchable electronics synthesized by vapor phase deposition. The second paper: "Diffusion of Metals in Polymers", Materials Science and Engineering, R22 1998, 5-5, [https://doi.org/10.1016/S0927-796X\(97\)00020-X](https://doi.org/10.1016/S0927-796X(97)00020-X) , by F. Faupel et al. already reports on detailed investigations emphasizing the key role of the deposition rate in nanostructure formation upon evaporation of Au and other noble metals onto polymers, also above the glass transition. Experiments and simulations are presented that show the transition from isolated nanoparticles at low deposition rates to a continuous film at high rates.

The group also published various follow-up papers on the subject.

In conclusion, except for the addition of the abovementioned papers, I recommend publication of the carefully revised manuscript in its present form. Proofreading is still required, e.g. on p10, line 230: "that that".

Replies to the Reviewers' Comments.

We would like to thank the reviewers for their careful reading of the revised manuscript and many valuable suggestions for improvements. We have considered the remarks and made all necessary changes to fully address the comments. The verbatim comments are given in black font. Our replies follow the corresponding comments in blue font.

Reviewer #1

Comment 1-1: “Kinetically Controlled Gold–Elastomer Nanophases for Environmentally Resilient Stretchable Electronics” by Soosang Chae, Won Jin Choi, Lisa Julia Nebel, Chang Hee Cho, Quinn A. Besford, André Knapp, Pavlo Makushko, Yevhen Zabala, Oleksandr Pylypovskyi, Min Woo Jeong, Stanislav Avdoshenko, Oliver Sander, Denys Makarov, Yoon Jang Chung, Andreas Fery, Jin Young Oh and Tae Il Lee.

I have carefully read the revised version of the manuscript NCHEM-23081735 transferred to Nature Communications. I am fully satisfied with the response of the authors to the points addressed in my earlier Reviewer Report to this manuscript and the changes made in the revised manuscript. I think that these changes have improved the readability and clarity of the manuscript. Hence, I recommend publication of the revised version of the manuscript in Nature Communications.

Response 1-1: We sincerely appreciate *Reviewer #1* for the positive and encouraging comments. The revised manuscript truly benefited from your comments and suggestions. Specifically, we have enhanced the readability by putting an emphasis on our core results and restructuring the materials.

Reviewer #2

Comment 2-1: The authors report on a method to obtain strain-invariant stretchable conductors by very slow evaporation of Au onto PDMS containing an excess of mobile crosslinker molecules. Based on extensive and careful analytical investigations and complementary simulations they developed a model for the reproducible formation of a buckled three-dimensional nanostructure which combines remarkable stretchability, strain-invariance, and

robustness under harsh environmental conditions. Key control parameters are the metal deposition rate and the amount of crosslinker.

Stretchable conductors are of strong interest, particularly for stretchable electronics and soft robotics, and have been investigated extensively. Some studies also report strain-invariant stretchable conductors. This feature is crucial in some applications. Nevertheless, the present work, from my point of view warrants publication in a high-impact journal like Nature Communications because, the reported approach is novel and, different from other approaches, it achieves the desired properties like high stretchability, strain-invariance, and resistance to harsh environment at the same time.

Response 2-1: We sincerely appreciate *Reviewer #2* for acknowledging key findings and assisting us in improving the manuscript.

Comment 2-2: The manuscript is well written and structured, and the comments of the reviewers have been adequately addressed. Concerning earlier work, I suggest to include two additional papers in the introductory part. In the paper: “Highly Deformable Nanostructured Elastomeric Electrodes With Improving Conductivity Upon Cyclical Stretching” G. Corbelli et al., first published: 29 August 2011, <https://doi.org/10.1002/adma.201102463>, the group of Paolo Milani reports on stretchable conductors based on Au nanostructures obtained by deposition of Au nanoclusters onto PDMS. This approach, which was refined in various follow-up papers, is different from the present one but should be mentioned in the introduction because the authors also report on Au/PDMS nanostructures for stretchable electronics synthesized by vapor phase deposition. The second paper: “Diffusion of Metals in Polymers”, Materials Science and Engineering, R22 1998, 5-5, [https://doi.org/10.1016/S0927-796X\(97\)00020-X](https://doi.org/10.1016/S0927-796X(97)00020-X) , by F. Faupel et al. already reports on detailed investigations emphasizing the key role of the deposition rate in nanostructure formation upon evaporation of Au and other noble metals onto polymers, also above the glass transition. Experiments and simulations are presented that show the transition from isolated nanoparticles at low deposition rates to a continuous film at high rates. The group also published various follow-up papers on the subject.

In conclusion, except for the addition of the abovementioned papers, I recommend publication of the carefully revised manuscript in its present form. Proofreading is still required, e.g. on p10, line 230: “that that”.

Response 2-2: Indeed, the references suggested by Reviewer #2 have been instrumental in shaping our concept of kinetically controlled metal-elastomer nanophases. We have incorporated these references into the introduction section. Additionally, we carefully reviewed the manuscript, addressing typos and errors. Thank you for bringing these to our attention.